# Stochastic Capital–Labor Lévy Jump Model with the Precariat Labor Force

Jaouad Danane 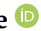

Laboratory of Systems Modelization and Analysis for Decision Support, National School of Applied Sciences, Hassan First University of Settat, Berrechid 26100, Morocco; jaouaddanane@gmail.com

**Abstract:** In this work, we study a capital–labor model by considering the interaction between the new proposed and the confirmed free jobs, the precariat labor force, and the mature labor force by introducing Brownian motion and Lévy noise. Moreover, we illustrate the well-posedness of the solution. In addition, we establish the conditions of the extinction of both the free jobs and labor force; subsequently, we prove the persistence of only the free jobs, and we also show the conditions of the persistence of both the free jobs and labor force. Finally, we validate our theoretical finding by numerical simulation by building a new stochastic Runge–Kutta method.

**Keywords:** capital–labor; persistence; extinction; precariat; Lévy jump; stochastic Runge–Kutta





## 1. Introduction

It is widely believed that the world's income inequality has increased over past decades [1,2]. Many different theories seek to explain this phenomenon, such as the evolution of the shares of capital–labor [3], changing returns to human capital and skill-driven technological change [4–7], and market concentration resulting from corporate market power and oligopoly [8,9]. Other work has concentrated more on the relationships between employers and workers and changing institutions [1,10–13]. Still others have focused on the potential changes to the economic class structure and the importance of economic class [1,14–16]. The evolution of institutions and the evolution of class structure can be seen as related rather than concurrent phenomena. The relation of workers to their work focused on a transformation of an old working class into a new working class, the precariat. The precariat is different from the old working class by its instability; they are a source of flexible labor, dependent on money wages due to a loss of labor rights and a weakened ability to access the welfare state, and without long-term stable employment [15,17,18]. In [19], Greenstein showed how the precarious class diagram can be applied empirically to data for the US labor force to help understand the changing nature of working-class jobs and who occupies those jobs. Moreover, he also explored to what extent this class structure can explain rising income inequality in the United States. Wage employment has also been on a downward trend in rich economies [20]. In [21], Breman et al. suggest that a trend of casualization in the rich world is inevitable due to a stagnant economy, a dismantled welfare state, and the decline of labor unions, among other causes.

The mathematical modeling of dynamics of capital–labor is an important tool to understand and study the behavior of interaction between free jobs and the labor force; for example, in [22], Riad et al. gave a deterministic model with logistic growth rate to established the condition of the existence and the stability of the equilibrium. More recently, in a stochastic study of the interaction between the free jobs and the labor force by introducing the white noise [23], also they found some conditions of the persistence and the extinction of labor force.

Motivated by the previous studies, we proposed the following capital–labor model with a Lévy jump:

$$
\begin{cases}
du_1(t) = & (\delta u_2(t) - \mu_1 u_1(t) - \kappa u_1(t))dt + \sigma_1 u_1(t)dW_1(t) + \displaystyle\int_U \Pi_1(u)u_1(t-)\tilde{N}(dt,du), \\[2mm]
du_2(t) = & \left(\kappa u_1(t) - \mu_2 u_2(t) - \tau_1 u_2^2(t) - \dfrac{cu_2 v_2}{1+\alpha u_2}\right)dt + \sigma_2 u_2(t)dW_2(t) \\[2mm]
& + \displaystyle\int_U \Pi_2(u)u_2(t-)\tilde{N}(dt,du), \\[2mm]
dv_1(t) = & \left(\dfrac{cu_2 v_2}{1+\alpha u_2} - \mu_3 v_1(t) - \theta v_1(t)\right)dt + \sigma_3 v_1(t)dW_3(t) \\[2mm]
& + \displaystyle\int_U \Pi_3(u)v_1(t-)\tilde{N}(dt,du), \\[2mm]
dv_2(t) = & \left(\theta v_1(t) - \mu_4 v_2(t) - \tau_2 v_2^2(t)\right)dt + \sigma_4 v_2(t)dW_4(t) + \displaystyle\int_U \Pi_4(u)v_2(t-)\tilde{N}(dt,du),
\end{cases}
\tag{1}
$$

where $u_1$ represents the new proposed jobs, $u_2$ the confirmed free jobs, $v_1$ the labor force in the training period and the precariat individuals (or the prematuration labor force), $v_2$ the maturation labor force, $\delta$ is the rate of new proposed free jobs, $\kappa$ is the rate of confirmed free jobs, $\mu_1$ is the loss rate of new free jobs, $\mu_2$ is the loss rate of confirmed free jobs, $\tau_1$ is the competition rate between confirmed free jobs, $\theta$ is the maturity rate of the immature labor force, $\mu_3$ is the loss rate of the jobs of the prematuration labor force, $\mu_4$ is the loss rate of the jobs of the mature labor force, $\tau_2$ is the competition rate between a mature labor force, $c$ is the rate that a labor force hunts a free job, and $\alpha$ is the effects of the recruitment rate on free jobs.

On a complete probability space $(\Omega, \mathcal{F}, (\mathcal{F}_t)_{t\geq0}, \mathbb{P})$, we define $W_i(t)$ as the standard Brownian motion, with the filtration $(\mathcal{F}_t)_{t\geq0}$ satisfying the classical conditions. Note that $u_1(t-)$, $u_2(t-)$, $v_1(t-)$, and $v_2(t-)$, respectively, are the left limits of $u_1(t)$, $u_2(t)$, $v_1(t)$, and $v_2(t)$; $\tilde{N}(dt,du) = N(dt,du) - \nu(du)dt$, with $N(dt,du)$ is a Poisson count measurement with the stationary compensator $\nu(du)dt$, and on a measurable subset $U$ of the positive half-line we define $\nu$ under the assumptions $\nu(U) < \infty$ and the intensity of $W_i(t)$ is $\sigma_i$, $q_i(u)$ represents the jump intensities for $i = 1, \ldots, 4$.

The organization of our study as follows. In the Section 2, we prove the well-posedness of the solution of the model (1). The extinction of both free jobs and the labor force is shown in Section 3. In Section 4, we show the persistence of free jobs and the extinction of the labor force. The stochastic persistence of both the free jobs and the labor force is studied in Section 5. Section 6 gives some numerical results in order to validate our theoretical findings.

## 2. Properties of the Solution

### 2.1. The Global Positive Solution Existence and Uniqueness

In this subsection, we will prove that the solution of the model in (1) exists and is unique.

**Theorem 1.** *If the initial value $(u_1(0), u_2(0), v_1(0), v_2(0)) \in \mathbb{R}_+^4$, then the model (1) admits a unique global solution $(u_1(t), u_2(t), v_1(t), v_2(t)) \in \mathbb{R}_+^4$ for $t \geq 0$ a.s.*

**Proof.** The diffusion and the drift are locally Lipschitz, so for any initial data $(u_1(0), u_2(0), v_1(0), v_2(0)) \in \mathbb{R}_+^4$, for $t \in [0, \tau_e)$ the system (1) admits a unique local solution $(u_1(t), u_2(t), v_1(t), v_2(t))$, where $\tau_e$ represents the explosion time.

For the purpose of proving that this solution is global, we must show that $\tau_e = \infty$ a.s. First, we demonstrate that for a finite time, $(u_1(t), u_2(t), v_1(t), v_2(t))$ do not tend to infinity. Let a sufficiently large number $n_0 > 0$, such that $(u_1(0), u_2(0), v_1(0), v_2(0))$ belong to the interval $[\frac{1}{n_0}, n_0]$. For each integer $n \geq n_0$, the stopping time is defined as follows:

$$
\tau_n = \inf\left\{t \in [0, \tau_e)/u_1(t) \notin \left(\frac{1}{n}, n\right) \text{ or } u_2(t) \notin \left(\frac{1}{n}, n\right) \text{ or } v_1(t) \notin \left(\frac{1}{n}, n\right) \text{ or } v_2(t) \notin \left(\frac{1}{n}, n\right)\right\},
$$

with $\tau_n$, when $n \uparrow \infty$, is an increasing number. Let $\tau_\infty = \lim_{n\to\infty} \tau_n$, where $\tau_\infty \leq \tau_e$ a.s. If we prove that $\tau_\infty = \infty$ then $\tau_e = \infty$ and $(u_1(t), u_2(t), v_1(t), v_2(t)) \in \mathbb{R}_+^4$ a.s. Suppose the contrary case is verified, i.e., $\tau_\infty < \infty$ a.s. Therefore, there are two constants $T > 0$ and $0 < \epsilon < 1$ such that $P(\tau_\infty \leq T) \geq \epsilon$.

Now, consider the functional

$$V(u_1(t), u_2(t), v_1(t), v_2(t)) = \sum_{i=1}^{i=2}(u_i - 1 - \log(u_i)) + \sum_{i=1}^{i=2}(v_i - 1 - \log(v_i)),$$

Using ItÔ's formula, we obtain

$$
\begin{aligned}
dV(x, y, z) =& LV \, dt + \sum_{i=1}^{i=2}\sigma_i(u_i - 1)\, dW_i + \sum_{i=1}^{i=2}\sigma_{i+2}(v_i - 1)\, dW_2 + \sigma_3(z-1)\, dW_{i+2} \\
&+ \int_U [\Pi_1(u)u_1 - \log(1 + \Pi_1(u))]\tilde{N}(dt, du) \\
&+ \int_U [\Pi_2(u)u_2 - \log(1 + \Pi_2(u))]\tilde{N}(dt, du) \\
&+ \int_U [\Pi_3(u)v_1 - \log(1 + \Pi_3(u))]\tilde{N}(dt, du) \\
&+ \int_U [\Pi_4(u)v_2 - \log(1 + \Pi_4(u))]\tilde{N}(dt, du),
\end{aligned}
\tag{2}
$$

with

$$
\begin{aligned}
LV =& \left(1 - \frac{1}{u_1}\right)(\delta u_2(t) - \mu_1 u_1(t) - \kappa u_1(t)) + \frac{\sigma_1^2}{2} \\
&+ \left(1 - \frac{1}{u_2}\right)\left(\kappa u_1(t) - \mu_2 u_2(t) - \tau_1 u_2^2(t) - \frac{c u_2 v_2}{1 + \alpha u_2}\right) + \frac{\sigma_2^2}{2} \\
&+ \left(1 - \frac{1}{v_1}\right)\left(\frac{c u_2 v_2}{1 + \alpha u_2} - \mu_3 v_1(t) - \theta v_1(t)\right) + \frac{\sigma_3^2}{2} \\
&+ \left(1 - \frac{1}{v_2}\right)\left(\theta v_1(t) - \mu_4 v_2(t) - \tau_2 v_2^2(t)\right) + \frac{\sigma_4^2}{2} \\
&+ \int_U [\Pi_1(u) - \log(1 + \Pi_1(u))]\nu(du) \\
&+ \int_U [\Pi_2(u) - \log(1 + \Pi_2(u))]\nu(du) \\
&+ \int_U [\Pi_3(u) - \log(1 + \Pi_3(u))]\nu(du) \\
&+ \int_U [\Pi_4(u) - \log(1 + \Pi_4(u))]\nu(du),
\end{aligned}
$$

then, we have

$$
\begin{aligned}
LV \leq& - \tau_1 u_2^2(t) + (\delta + \tau_1 + c)u_2 + p + \mu_1 + \mu_2 + \mu_3 + \mu_4 - \tau_2 v_2^2 + (c + \tau_2)v_2 + 4C' \\
LV \leq& \frac{(\delta + \tau_1 + c)^2}{4\tau_1} + \kappa + \mu_1 + \mu_2 + \mu_3 + \mu_4 + \frac{(\tau_2 + c)^2}{4\tau_2} + 4C' \\
LV \leq& C,
\end{aligned}
$$

with

$$
\begin{aligned}
C' = \max\bigg\{ &\int_U \Pi_1(u) - \log(1 + \Pi_1(u))\nu(du), \int_U (\Pi_2(u) - \log(1 + \Pi_2(u)))\nu(du), \\
&\int_U (\Pi_3(u) - \log(1 + \Pi_3(u)))\nu(du), \int_U (\Pi_4(u) - \log(1 + \Pi_4(u)))\nu(du)\bigg\}.
\end{aligned}
$$

and

$$C = \frac{(\delta + \tau_1 + c)^2}{4\tau_1} + \kappa + \mu_1 + \mu_2 + \mu_3 + \mu_4 + \frac{(\tau_2 + c)^2}{4\tau_2} + 4C'.$$

Then, Equation (2) implies that

$$0 \leq \mathbb{E}(V(u_1(\tau_n \wedge T), u_2(\tau_n \wedge T), v_1(\tau_n \wedge T), v_2(\tau_n \wedge T)))$$
$$\leq V(u_1(0), u_2(0), v_1(0), v_2(0)) + CT.$$

For any $h > 0$, we define

$$H(h) = \inf\{V(x_1, x_2, x_3, x_4), x_i \geq h \text{ or } x_i \leq \frac{1}{h}, \ i = 1, \ldots, 4\},$$

where $x_1 = u_1$, $x_2 = u_2$, $x_3 = v_1$ and $x_4 = v_2$. Therefore, we obtain

$$\lim_{h \to \infty} H(h) = \infty.$$

Then, letting $n \to \infty$, we find

$$\infty > V(u_1(0), u_2(0), v_1(0), v_2(0)) + CT = \infty,$$

which is a contradiction with the previous assumption. Therefore $\tau_\infty = \infty$. Moreover, our model admits a unique global solution $(u_1(t), u_2(t), v_1(t), v_2(t))$ a.s. □

### 2.2. Stochastic Ultimately Boundedness

In the previous subsection, we have proven that the solution of the model (1) is positive. Nevertheless, this non explosion property in a dynamical system is often insufficient. Therefore, the stochastic ultimate boundedness is more desired.

**Theorem 2.** *If we have the following conditions*

$$\sigma_1^2 + \int_U \Pi_1^2(u)\nu(du) - 2\mu_1 - 2\kappa + 1 < 0,$$
$$\sigma_2^2 + \int_U \Pi_2^2(u)\nu(du) - 2\mu_2 + 1 < 0,$$
$$\sigma_3^2 + \int_U \Pi_3^2(u)\nu(du) - 2\mu_3 - 2\theta + 1 < 0,$$
$$\sigma_4^2 + \int_U \Pi_4^2(u)\nu(du) - 2\mu_4 + 1 < 0.$$

(3)

*Then, for an initial data $(u_1(0), u_2(0), v_1(0), v_2(0)) \in \mathbb{R}_+^4$, the solution of model (1) is stochastic ultimately bounded.*

**Proof.** Define the following function:

$$V(u_1, u_2, v_1, v_2) = \sum_{i=1}^{i=2} u_i^2 + \sum_{i=1}^{i=2} v_i^2$$

By ItÔ's formula, we get

$$dV(u_1, u_2, v_1, v_2) = LV \, dt + 2\sigma_1^2 u_1^2 dW_1(t) + 2\sigma_2^2 u_2^2 dW_2(t) + 2\sigma_3^2 v_1^2 dW_3(t) + 2\sigma_4^2 v_2^2 dW_4(t) \tag{4}$$

$$+ \int_U (2\Pi_1(u) + \Pi_1^2(u)) u_1^2 \tilde{N}(dt, du) + \int_U (2\Pi_2(u) + \Pi_2^2(u)) u_2^2 \tilde{N}(dt, du) \tag{5}$$

$$+ \int_U (2\Pi_3(u) + \Pi_3^2(u)) v_1^2 \tilde{N}(dt, du) + \int_U (2\Pi_4(u) + \Pi_4^2(u)) v_2^2 \tilde{N}(dt, du), \tag{6}$$

with

$$
\begin{aligned}
LV = {}& 2u_1(\delta u_2 - \mu_1 u_1 - \kappa u_1) + 2u_2\left(\kappa u_1 - \mu_2 u_2 - \tau_1 u_2^2 - \frac{cu_2 v_2}{1 + \alpha u_2}\right) \\
& + 2v_1\left(\frac{cu_2 v_2}{1 + \alpha u_2} - \mu_3 v_1 - \theta v_1\right) + 2v_2\left(\theta v_1 - \mu_4 v_2 - \tau_2 v_2^2\right) \\
& + \sigma_1 u_1^2 + \sigma_2 u_2^2 + \sigma_3 v_1^2 + \sigma_4 v_2^2 + \int_U \Pi_1^2 u_1^2 \nu(du) \\
& + \int_U \Pi_2^2 u_2^2 \nu(du) + \int_U \Pi_3^2 v_1^2 \nu(du) + \int_U \Pi_4^2 v_2^2 \nu(du),
\end{aligned}
$$

so,

$$
\begin{aligned}
LV = {}& -2\tau_1 u_2^3 - 2\tau_2 v_2^3 + \left(\sigma_1^2 + \int_U \Pi_1^2(u)\nu(du) - 2\mu_1 - 2\kappa + 1\right) u_1^2 \\
& + \left(\sigma_2^2 + \int_U \Pi_2^2(u)\nu(du) - 2\mu_2 + 1\right) u_2^2 + \left(\sigma_3^2 + \int_U \Pi_3^2(u)\nu(du) - 2\mu_3 - 2\theta + 1\right) v_1^2 \\
& + \left(\sigma_4^2 + \int_U \Pi_4^2(u)\nu(du) - 2\mu_4 + 1\right) v_1^2 + 2(a + p)u_1 u_2 + 2\theta v_1 v_2 \\
& + \frac{2cu_2 v_1 v_2}{1 + \alpha u_2} - \frac{2cu_2^2 v_2}{1 + \alpha u_2} - \left(u_1^2 + u_2^2 + v_1^2 + v_2^2\right),
\end{aligned}
$$

then,

$$
\begin{aligned}
LV \leq {}& \left(\sigma_1^2 + \int_U \Pi_1^2(u)\nu(du) - 2\mu_1 - 2\kappa + 1\right) u_1^2 + \left(\sigma_2^2 + \int_U \Pi_2^2(u)\nu(du) - 2\mu_2 + 1\right) u_2^2 \\
& + \left(\sigma_3^2 + \int_U \Pi_3^2(u)\nu(du) - 2\mu_3 - 2\theta + 1\right) v_1^2 + \left(\sigma_4^2 + \int_U \Pi_4^2(u)\nu(du) - 2\mu_4 + 1\right) v_2^2 \\
& + 2(a + p)u_1 u_2 + 2(h + c)v_1 v_2 - \left(u_1^2 + u_2^2 + v_1^2 + v_2^2\right),
\end{aligned}
$$

we denote

$$
\begin{aligned}
f(u_1, u_2, v_1, v_2) = {}& \left(\sigma_1^2 + \int_U \Pi_1^2(u)\nu(du) - 2\mu_1 - 2\kappa + 1\right) u_1^2 + \left(\sigma_2^2 + \int_U \Pi_2^2(u)\nu(du) - 2\mu_2 + 1\right) u_2^2 \\
& + \left(\sigma_3^2 + \int_U \Pi_3^2(u)\nu(du) - 2\mu_3 - 2\theta + 1\right) v_1^2 + \left(\sigma_4^2 + \int_U \Pi_4^2(u)\nu(du) - 2\mu_4 + 1\right) v_2^2 \\
& + 2(a + p)u_1 u_2 + 2(h + c)v_1 v_2.
\end{aligned}
$$

Since we have condition (3), we obtain that the function $f(u_1, u_2, v_1, v_2)$ admits an upper bound. We put

$$M = \sup_{(u_1, u_2, v_1, v_2) \in \mathbb{R}_+^4} f(u_1, u_2, v_1, v_2) \text{ and } L = M + 1.$$

Since $f(0, 0, 0, 0) = 0$, then $L > 0$. Formula (4) implies that

$$dV \leq \left[ L - (u_1^2 + u_2^2 + v_1^2 + v_2^2) \right] dt + 2\sigma_1^2 u_1^2 dW_1(t) + 2\sigma_2^2 u_2^2 dW_2(t) + 2\sigma_3^2 v_1^2 dW_3(t) + 2\sigma_4^2 v_2^2 dW_4(t)$$
$$+ \int_U (2\Pi_1(u) + \Pi_1^2(u)) u_1^2 \tilde{N}(dt, du) + \int_U (2\Pi_2(u) + \Pi_2^2(u)) u_2^2 \tilde{N}(dt, du)$$
$$+ \int_U (2\Pi_3(u) + \Pi_3^2(u)) v_1^2 \tilde{N}(dt, du) + \int_U (2\Pi_4(u) + \Pi_4^2(u)) v_2^2 \tilde{N}(dt, du),$$

therefore, we use ItÔ's formula to get

$$
\begin{aligned}
d\left[ e^t V \right] =& e^t V \, dt + e^t dV \\
\leq& N e^t \, dt + 2\sigma_1^2 u_1^2 dW_1(t) + 2\sigma_2^2 u_2^2 dW_2(t) + 2\sigma_3^2 v_1^2 dW_3(t) + 2\sigma_2^2 v_2^2 dW_4(t) \\
&+ \int_U (2\Pi_1(u) + \Pi_1^2(u)) u_1^2 \tilde{N}(dt, du) + \int_U (2\Pi_2(u) + \Pi_2^2(u)) u_2^2 \tilde{N}(dt, du) \\
&+ \int_U (2\Pi_3(u) + \Pi_3^2(u)) v_1^2 \tilde{N}(dt, du) + \int_U (2\Pi_4(u) + \Pi_4^2(u)) v_2^2 \tilde{N}(dt, du),
\end{aligned}
\tag{7}
$$

then,

$$e^t \mathbb{E}[V(X)] \leq V(u_1(0), u_2(0), v_1(0), v_2(0)) + Le^t - L.$$

with $X = (u_1, u_2, v_1, v_2)$.

Therefore

$$\limsup_{t \to \infty} \mathbb{E}[V(X)] \leq L.$$

We know that $V(X) = \sum_{i=1}^{i=2} u_i^2 + \sum_{i=1}^{i=2} v_i^2$, then

$$\limsup_{t \to \infty} \mathbb{E}[|(X)|^2] \leq L.$$

By Chebyshev's inequality, for any $\eta > 0$, let $B = \dfrac{\sqrt{L}}{\sqrt{\eta}}$. we get

$$P(|(X)| > B) \leq \frac{\mathbb{E}[|(X)|^2]}{B^2} \leq \frac{L}{\frac{L}{\eta}} = \eta.$$

□

## 3. Stochastic Extinction of Free Jobs and the Labor Force

Now, we prove that free jobs and the labor force becomes extinct with probability one. We put

$$
\begin{aligned}
c_1 &= 2\mu_1 + p - \sigma_1^2 - \int_U \Pi_1^2(u)\nu(du), \\
c_2 &= 2\mu_2 - \sigma_2^2 - \int_U \Pi_2^2(u)\nu(du) - \delta - \kappa, \\
c_3 &= 2\mu_3 + h - \sigma_3^2 - \int_U \Pi_3^2(u)\nu(du) - \frac{c}{\alpha}, \\
c_4 &= 2\mu_4 - \sigma_4^2 - \int_U \Pi_4^2(u)\nu(du) - \frac{c}{\alpha} - \theta.
\end{aligned}
$$

**Theorem 3.** *For any initial data* $(u_1(0), u_2(0), v_1(0), v_2(0)) \in \mathbb{R}_+^4$, *we obtain*

$$\limsup_{t \to \infty} \frac{1}{t} \mathbb{E}\left(+u_1^2 + u_2^2 + v_1^2 + v_2^2\right) \leq \frac{L_1}{\varrho},$$

*when,* $\min\{c_1, c_2, c_3, c_4\} > 0$.

**Proof.** We define

$$F(u_1, u_2, v_1, v_2) = \sum_{i=1}^{i=2} u_i^2 + \sum_{i=1}^{i=2} v_i^2 + u_1 + u_2 + v_1,$$

By ItÔ's formula, we get

$$
\begin{aligned}
dF =& LF\,dt + \sigma_1^2(2u_1^2 + u_1)dW_1(t) + \sigma_2^2(2u_2^2 + u_2)dW_2(t) + \sigma_3^2(2v_1^2 + v_1)dW_3(t) \\
&+ 2\sigma_4^2 v_2^2 dW_4(t) + \int_U (2\Pi_1(u) + \Pi_1^2(u))u_1^2 + \Pi_1 u_1 \tilde{N}(dt, du) \\
&+ \int_U (2\Pi_2(u) + \Pi_2^2(u))u_2^2 + \Pi_2 u_2 \tilde{N}(dt, du) + \int_U (2\Pi_3(u) + \Pi_3^2(u))v_1^2 + \Pi_3 v_1 \tilde{N}(dt, du) \\
&+ \int_U (2\Pi_4(u) + \Pi_4^2(u))v_2^2 \tilde{N}(dt, du),
\end{aligned}
\tag{8}
$$

where

$$
\begin{aligned}
LF =& 2u_1(\delta u_2 - \mu_1 u_1 - \kappa u_1) + 2u_2\left(\kappa u_1 - \mu_2 u_2 - \tau_1 u_2^2 - \frac{cu_2 v_2}{1 + \alpha u_2}\right) \\
&+ 2v_1\left(\frac{cu_2 v_2}{1 + \alpha u_2} - \mu_3 v_1 - hv_1\right) + 2v_2\left(\theta v_1 - \mu_4 v_2 - \tau_2 v_2^2\right) \\
&+ \delta u_2 - \tau_1 u_2^2 - \mu_1 u_1 - \mu_2 u_2 - (\mu_3 + \theta)v_1 + \sigma_1 u_1^2 + \sigma_2 u_2^2 + \sigma_3 v_1^2 \\
&+ \sigma_4 v_2^2 + \int_U \Pi_1^2 u_1^2 \nu(du) + \int_U \Pi_2^2 u_2^2 \nu(du) + \int_U \Pi_3^2 v_1^2 \nu(du) + \int_U \Pi_4^2 v_2^2 \nu(du),
\end{aligned}
$$

so,

$$
\begin{aligned}
LF =& -2\tau_1 u_2^3 - 2\tau_2 v_2^3 + \left(\sigma_1^2 + \int_U \Pi_1^2(u)\nu(du) - 2\mu_1 - 2\kappa\right)u_1^2 \\
&+ \left(\sigma_2^2 + \int_U \Pi_2^2(u)\nu(du) - 2\mu_2\right)u_2^2 + \left(\sigma_3^2 + \int_U \Pi_3^2(u)\nu(du) - 2\mu_3 - 2\theta\right)v_1^2 \\
&+ \left(\sigma_4^2 + \int_U \Pi_4^2(u)\nu(du) - 2\mu_4\right)v_1^2 + 2(\delta + \kappa)u_1 u_2 + 2\theta v_1 v_2 - (\mu_3 + \theta)v_1 \\
&+ \frac{2cu_2 v_1 v_2}{1 + \alpha u_2} - \frac{2cu_2^2 v_2}{1 + \alpha u_2} + \delta u_2 - \tau_1 u_2^2 - \mu_1 u_1 - \mu_2 u_2,
\end{aligned}
$$

then,

$$
\begin{aligned}
LF \leq& \left(\sigma_1^2 + \int_U \Pi_1^2(u)\nu(du) - 2\mu_1 - 2\kappa\right)u_1^2 \\
&+ \left(\sigma_2^2 + \int_U \Pi_2^2(u)\nu(du) - 2\mu_2\right)u_2^2 + \left(\sigma_3^2 + \int_U \Pi_3^2(u)\nu(du) - 2\mu_3 - 2\theta\right)v_1^2 \\
&+ \left(\sigma_4^2 + \int_U \Pi_4^2(u)\nu(du) - 2\mu_4\right)v_1^2 + (\delta + \kappa)u_1^2 + (\delta + \kappa)u_2^2 + \theta v_1^2 + \theta v_2^2 \\
&+ \frac{c}{\alpha}v_1^2 + \frac{c}{\alpha}v_2^2 + \frac{a}{4\tau_1},
\end{aligned}
$$

this fact implies that,

$$LF \leq \left( \sigma_1^2 + \int_U \Pi_1^2(u)\nu(du) - 2\mu_1 - \kappa \right) u_1^2$$

$$+ \left( \sigma_2^2 + \int_U \Pi_2^2(u)\nu(du) + \delta + \kappa - 2\mu_2 \right) u_2^2 + \left( \sigma_3^2 + \int_U \Pi_3^2(u)\nu(du) + \frac{c}{\alpha} - 2\mu_3 - \theta \right) v_1^2$$

$$+ \left( \sigma_4^2 + \int_U \Pi_4^2(u)\nu(du) + \frac{c}{\alpha} + \theta - 2\mu_4 \right) v_1^2 + L_1,$$

with $L_1 = \dfrac{\delta}{4\tau_1}$. Then

$$LF \leq - c_1 u_1^2 - c_2 u_2^2 - c_3 v_1^2 - c_4 v_2^2 + L_1,$$

Equation (8) implies that

$$0 \leq \mathbb{E}(F(u_1(t), u_2(t), v_1(t), v_2(t))) \leq \mathbb{E}\left\{ \int_0^t \left( -c_1 u_1^2 - c_2 u_2^2 - c_3 v_1^2 - c_4 v_2^2 \right) \right\} + L_1 t$$

$$+ F(u_1(0), u_2(0), v_1(0), v_2(0))$$

let $\varrho = \min\{c_1, c_2, c_3, c_4\}$, so

$$\mathbb{E}\left\{ \int_0^t \left( u_1^2 + u_2^2 + v_1^2 + v_2^2 \right) \right\} \leq \frac{L_1}{\varrho} t + \frac{(u_1(0), u_2(0), v_1(0), v_2(0))}{\varrho}.$$

Then,

$$\limsup_{t \to \infty} \frac{1}{t} \mathbb{E}\left\{ \int_0^t \left( u_1^2 + u_2^2 + v_1^2 + v_2^2 \right) \right\} \leq \frac{L_1}{\varrho}.$$

□

## 4. Stochastic Extinction of Capital Labor

In this section, we established some conditions to proving the extinction of the labor force with probability one.

First, we denote

$$c_3 = 2\mu_3 + \theta - \sigma_3^2 - \int_U \Pi_3^2(u)\nu(du) - \frac{c}{\alpha},$$

$$c_4 = 2\mu_4 - \tau_2 - \sigma_4^2 - \int_U \Pi_4^2(u)\nu(du) - \frac{c}{\alpha} - h,$$

$$N_1 = \sum_{i=1}^{i=2} \frac{\sigma_i^2}{2} + 2N_1' - (\mu_1 + \kappa + \mu_2),$$

where

$$N_1' = \min\left\{ \int_U \Pi_1(u) - \ln(1 + \Pi_1(u)), \int_U \Pi_2(u) - \ln(1 + \Pi_2(u)) \right\}.$$

**Theorem 4.** *For any initial data* $(u_1(0), u_2(0), v_1(0), v_2(0)) \in \mathbb{R}_+^4$, *when* $\min\{c_3, lc_4, N_1\} > 0$ *and* $\dfrac{v_2}{u_1} \leq \dfrac{\alpha p}{c}$, *we obtain*

$$\limsup_{t \to \infty} \frac{1}{t} \mathbb{E}\left( v_1^2 + v_2^2 \right) \leq \frac{L_2}{\varrho_1}$$

*then, all the prey persistence in the mean. Moreover, we have*

$$\frac{1}{t}\int_0^t u_2(\eta)d\eta \geq \frac{N_1}{\tau_1} \ a.s.$$

$$\frac{1}{t}\int_0^t u_1(\eta)d\eta \geq \frac{\delta N_1}{\tau_1(\mu_1+\kappa)} \ a.s.$$

**Proof.** We consider

$$F_1(v_1,v_2) = \sum_{i=1}^{i=2} v_i^2 + v_2,$$

by ItÔ's formula, we obtain

$$
\begin{aligned}
dF_1(v_1,v_2) =& LF_1 \ dt + \sigma_3^2(2v_1^2+v_1)dW_3(t) + 2\sigma_4^2 v_2^2 dW_4(t) \\
&+ \int_U (2\Pi_3(u)+\Pi_3^2(u))v_1^2 + \Pi_3 v_1 \tilde{N}(dt,du) + \int_U (2\Pi_4(u)+\Pi_4^2(u))v_2^2 \tilde{N}(dt,du),
\end{aligned}
\tag{9}
$$

with

$$
\begin{aligned}
LF_1 =& 2v_1\left(\frac{cu_2 v_2}{1+\alpha u_2} - \mu_3 v_1 - \theta v_1\right) + 2v_2\left(\theta v_1 - \mu_4 v_2 - \tau_2 v_2^2\right) \\
&+ \frac{cu_2 v_2}{1+\alpha u_2} - (\mu_3+h)v_2 \\
&+ \sigma_3 v_1^2 + \sigma_4 v_2^2 + \int_U \Pi_3^2 v_1^2 \nu(du) + \int_U \Pi_4^2 v_2^2 \nu(du),
\end{aligned}
$$

so,

$$
\begin{aligned}
LF_1 =& -2\tau_2 v_2^3 + \left(\sigma_3^2 + \int_U \Pi_3^2(u)\nu(du) - 2\mu_3 - 2\theta\right)v_1^2 \\
&+ \left(\sigma_4^2 + \int_U \Pi_4^2(u)\nu(du) - 2\mu_4\right)v_1^2 + 2\theta v_1 v_2 \\
&+ \frac{2cu_2 v_1 v_2}{1+\alpha u_2} - \frac{cu_2 v_2}{1+\alpha u_2} - (\mu_3+\theta)v_2,
\end{aligned}
$$

Therefore,

$$
\begin{aligned}
LF_1 \leq& \left(\sigma_3^2 + \int_U \Pi_3^2(u)\nu(du) - 2\mu_3 - 2\theta\right)v_1^2 \\
&+ \left(\sigma_4^2 + \int_U \Pi_4^2(u)\nu(du) - 2\mu_4\right)v_1^2 + hv_1^2 + \tau_2 v_2^2 \\
&+ \frac{c}{\alpha}v_1^2 + \frac{c}{\alpha}v_2^2 - (\mu_3+\theta)v_2,
\end{aligned}
$$

then,

$$
\begin{aligned}
LF_1 \leq& \left(\sigma_3^2 + \int_U \Pi_3^2(u)\nu(du) + \frac{c}{\alpha} - 2\mu_3 - \theta\right)v_1^2 \\
&+ \left(\sigma_4^2 + \int_U \Pi_4^2(u)\nu(du) + \tau_2 - 2\mu_4\right)v_1^2 + L_2,
\end{aligned}
$$

with $L_2 = \dfrac{\alpha(\mu_3+\theta-\frac{c}{\alpha})^2}{4c}$. Therefore,

$$LF_1 \leq -c_3 v_1^2 - c_4 v_2^2 + L_2,$$

Using Equation (9), we obtain

$$0 \leq \mathbb{E}(F_1(v_1(t), v_2(t))) \leq \mathbb{E}\left\{ \int_0^t \left( -c_3 v_1^2 - c_4 v_2^2 \right) \right\} + L_2 t + F_1(v_1(0), v_2(0)),$$

Noting $\varrho_1 = \min\{c_3, c_4\}$, so

$$\mathbb{E}\left\{ \int_0^t \left( v_1^2 + v_2^2 \right) \right\} \leq \frac{L_2}{\varrho_1} t + \frac{F_1(v_1(0), v_2(0))}{\varrho_1},$$

then,

$$\limsup_{t \to \infty} \frac{1}{t} \mathbb{E}\left\{ \int_0^t \left( v_1^2 + v_2^2 \right) \right\} \leq \frac{L_2}{\varrho_1}$$

Let

$$G_1(u_1, u_2) = \ln(u_1 u_2),$$

by ItÔ's formula, we get

$$dG_1 = LG_1 \, dt + \sum_{i=1}^{i=2} \left( \sigma_i \, dW_i + \int_U q_i(u) u_i - \ln(1 + q_i(u)) \tilde{N}(dt, du) \right),$$

with

$$LG_1 = \frac{\delta u_2}{u_1} - (\mu_1 + \kappa) + \frac{\kappa u_1}{u_2} - \mu_2 - \tau_1 u_2 - \frac{c v_2}{1 + \alpha u_2}$$
$$+ \sum_{i=1}^{i=2} \left( \frac{\sigma_i^2}{2} + \int_U q_i(u) - \ln(1 + q_i(u)) \nu(du) \right).$$

We know $\dfrac{v_2}{u_1} \leq \dfrac{\alpha \kappa}{c}$. Therefore,

$$LG_1 \geq -\tau_1 u_2 - (\mu_1 + \kappa + \mu_2) + \sum_{i=1}^{i=2} \frac{\sigma_i^2}{2} + 2N_1'$$

where $N_1' = \min\left\{ \int_U \Pi_1(u) - \ln(1 + \Pi_1(u)), \int_U \Pi_2(u) - \ln(1 + \Pi_2(u)) \right\}$, so

$$LG_1 \geq -\tau_1 u_2 + N_1$$

with

$$N_1 = \sum_{i=1}^{i=2} \frac{\sigma_i^2}{2} + 2N_1' - (\mu_1 + \kappa + \mu_2).$$

Therefore,

$$\ln(u_1(t) u_2(t)) \geq \ln(u_1(0) u_2(0)) - \tau_1 \int_0^t u_2(\eta) d\eta + K_1 t$$
$$+ \sum_{i=1}^{i=2} \int_0^t \left( \sigma_i \, dW_i(\eta) + \int_U q_i(u) u_i - \ln(1 + q_i(u)) \tilde{N}(d\eta, du) \right).$$

Therefore,

$$\frac{\ln(u_1(t)u_2(t))}{t} \geq \frac{\ln(u_1(0)u_2(0))}{t} - \tau_1 \frac{1}{t}\int_0^t u_2(\eta)d\eta + K_1$$
$$+ \sum_{i=1}^{i=2} \frac{1}{t}\int_0^t \left( \sigma_i \, dW_i(\eta) + \int_U q_i(u)u_i - \ln(1+q_i(u))\tilde{N}(d\eta, du) \right).$$

The strong law of large numbers for local martingales implies that

$$0 \geq -\tau_1 \frac{1}{t}\int_0^t u_2(\eta)d\eta + N_1,$$

then,

$$\frac{1}{t}\int_0^t u_2(\eta)d\eta \geq \frac{N_1}{\tau_1} \text{ a.s.}$$

The first equation of system (1) implies that

$$du_1(t) = (\delta u_2(t) - \mu_1 u_1(t) - \kappa u_1(t))dt + \sigma_1 u_1(t)dW_1(t) + \int_U \Pi_1(u)u_1(t-)\tilde{N}(dt, du),$$

so,

$$u_1(t) - u_1(0) = \delta \int_0^t u_2(\eta)d\eta - (\mu_1 + \kappa)\int_0^t u_1(\eta)d\eta + \int_0^t \sigma_1 u_1(\eta)dW_1(\eta)$$
$$+ \int_0^t \int_U \Pi_1(u)u_1(\eta-)\tilde{N}(d\eta, du),$$

then,

$$\frac{u_1(t) - u_1(0)}{t} = \delta \frac{1}{t}\int_0^t u_2(\eta)d\eta - (\mu_1 + \kappa)\frac{1}{t}\int_0^t u_1(\eta)d\eta + \frac{1}{t}\int_0^t \sigma_1 u_1(\eta)dW_1(\eta)$$
$$+ \frac{1}{t}\int_0^t \int_U \Pi_1(u)u_1(\eta-)\tilde{N}(d\eta, du).$$

Since $\frac{1}{t}\int_0^t u_2(\eta)d\eta \geq \frac{N_1}{\tau_1}$, so

$$\frac{u_1(t) - u_1(0)}{t} \geq \frac{\delta N_1}{\tau_1} - (\mu_1 + \kappa)\frac{1}{t}\int_0^t u_1(\eta)d\eta + \frac{1}{t}\int_0^t \sigma_1 u_1(\eta)dW_1(\eta)$$
$$+ \frac{1}{t}\int_0^t \int_U \Pi_1(u)u_1(\eta-)\tilde{N}(d\eta, du),$$

using the strong law of large numbers for local martingales, we get

$$0 \geq \frac{\delta N_1}{\tau_1} - (\mu_1 + \kappa)\frac{1}{t}\int_0^t u_1(\eta)d\eta,$$

then,

$$\frac{1}{t}\int_0^t u_1(\eta)d\eta \geq \frac{\delta N_1}{\tau_1(\mu_1 + \kappa)} \text{ a.s.}$$

$\square$

## 5. Stochastic Persistence

In order to prove the stochastic persistence of both the free jobs and the capital labor, we define the following constant

$$N_1 = \sum_{i=1}^{i=2} \frac{\sigma_i^2}{2} + 2N_1' - (\mu_1 + \kappa + \mu_2),$$

$$N_2 = \sum_{i=3}^{i=4} \frac{\sigma_i^2}{2} + 2N_2' - (\mu_3 + \theta + \mu_4),$$

with

$$N_1' = \min \left\{ \int_U \Pi_1(u) - \ln(1 + \Pi_1(u)), \int_U \Pi_2(u) - \ln(1 + \Pi_2(u)) \right\}$$

and

$$N_2' = \min \left\{ \int_U \Pi_3(u) - \ln(1 + \Pi_3(u)), \int_U \Pi_4(u) - \ln(1 + \Pi_4(u)) \right\}.$$

**Theorem 5.** *If $\min\{N_1, N_2\} > 0$ and $\frac{v_2}{u_1} \leq \frac{\alpha\kappa}{c}$, then, all the free jobs and labor force persistence in the mean. In addition, we have*

$$\frac{1}{t} \int_0^t u_2(\eta) d\eta \geq \frac{N_1}{\tau_1} \ a.s.$$

$$\frac{1}{t} \int_0^t u_1(\eta) d\eta \geq \frac{\delta N_1}{\tau_1(\mu_1 + \kappa)} \ a.s.$$

$$\frac{1}{t} \int_0^t v_2(\eta) d\eta \geq \frac{N_2}{\tau_2} \ a.s.$$

$$\frac{1}{t} \int_0^t v_1(\eta) d\eta \geq \frac{\mu_4 N_2}{\theta \tau_2} \ a.s.$$

**Proof.** We consider the following function:

$$G_1(u_1, u_2) = \ln(u_1 u_2),$$

ItÔ's formula implies that

$$dG_1 = LG_1 \, dt + \sum_{i=1}^{i=2} \left( \sigma_i \, dW_i + \int_U q_i(u) u_i - \ln(1 + q_i(u)) \tilde{N}(dt, du) \right),$$

with

$$LG_1 = \frac{\delta u_2}{u_1} - (\mu_1 + \kappa) + \frac{\kappa u_1}{u_2} - \mu_2 - \tau_1 u_2 - \frac{c v_2}{1 + \alpha u_2}$$
$$+ \sum_{i=1}^{i=2} \left( \frac{\sigma_i^2}{2} + \int_U q_i(u) - \ln(1 + q_i(u)) v(du) \right).$$

Since $\frac{v_2}{u_1} \leq \frac{\alpha\kappa}{c}$, then,

$$LG_1 \geq -\tau_1 u_2 - (\mu_1 + \kappa + \mu_2) + \sum_{i=1}^{i=2} \frac{\sigma_i^2}{2} + 2N_1'$$

where $N_1' = \min\left\{ \int_U \Pi_1(u) - \ln(1 + \Pi_1(u)), \int_U \Pi_2(u) - \ln(1 + \Pi_2(u)) \right\}$, then

$$LG_1 \geq -\tau_1 u_2 + N_1$$

with

$$N_1 = \sum_{i=1}^{i=2} \frac{\sigma_i^2}{2} + 2N_1' - (\mu_1 + \kappa + \mu_2).$$

Then,

$$\ln(u_1(t)u_2(t)) \geq \ln(u_1(0)u_2(0)) - \tau_1 \int_0^t u_2(\eta)d\eta + N_1 t$$
$$+ \sum_{i=1}^{i=2} \int_0^t \left( \sigma_i \, dW_i(\eta) + \int_U q_i(u)u_i - \ln(1 + q_i(u))\tilde{N}(d\eta, du) \right).$$

Thus fact implies that

$$\frac{\ln(u_1(t)u_2(t))}{t} \geq \frac{\ln(u_1(0)u_2(0))}{t} - \tau_1 \frac{1}{t} \int_0^t u_2(\eta)d\eta + N_1$$
$$+ \sum_{i=1}^{i=2} \frac{1}{t} \int_0^t \left( \sigma_i \, dW_i(\eta) + \int_U q_i(u)u_i - \ln(1 + q_i(u))\tilde{N}(d\eta, du) \right)$$

Using the strong law of large numbers for local martingales, we get

$$0 \geq -\tau_1 \frac{1}{t} \int_0^t u_2(\eta)d\eta + N_1,$$

therefore,

$$\frac{1}{t} \int_0^t u_2(\eta)d\eta \geq \frac{N_1}{\tau_1} \text{ a.s.}$$

Using the first equation of system (1), we get

$$u_1(t) - u_1(0) = \delta \int_0^t u_2(\eta)d\eta - (\mu_1 + \kappa) \int_0^t u_1(\eta)d\eta + \int_0^t \sigma_1 u_1(\eta)dW_1(\eta)$$
$$+ \int_0^t \int_U \Pi_1(u)u_1(\eta-)\tilde{N}(d\eta, du),$$

then,

$$\frac{u_1(t) - u_1(0)}{t} = \delta \frac{1}{t} \int_0^t u_2(\eta)d\eta - (\mu_1 + \kappa)\frac{1}{t} \int_0^t u_1(\eta)d\eta + \frac{1}{t} \int_0^t \sigma_1 u_1(\eta)dW_1(\eta)$$
$$+ \frac{1}{t} \int_0^t \int_U \Pi_1(u)u_1(\eta-)\tilde{N}(d\eta, du),$$

we know that $\frac{1}{t} \int_0^t u_2(\eta)d\eta \geq \frac{N_1}{\tau_1}$, so

$$\frac{u_1(t) - u_1(0)}{t} \geq \frac{\delta K_1}{\tau_1} - (\mu_1 + \kappa)\frac{1}{t} \int_0^t u_1(\eta)d\eta + \frac{1}{t} \int_0^t \sigma_1 u_1(\eta)dW_1(\eta)$$
$$+ \frac{1}{t} \int_0^t \int_U \Pi_1(u)u_1(\eta-)\tilde{N}(d\eta, du),$$

The strong law of large numbers for local martingales implies that

$$0 \geq \frac{\delta N_1}{\tau_1} - (\mu_1 + \kappa)\frac{1}{t}\int_0^t u_1(\eta)d\eta,$$

then,

$$\frac{1}{t}\int_0^t u_1(\eta)d\eta \geq \frac{\delta N_1}{\tau_1(\mu_1 + \kappa)} \text{ a.s.}$$

We use the following function:

$$G_2(v_1, v_2) = \ln(v_1 v_2),$$

ItÔ's formula implies that

$$dG_2 = LG_2\, dt + \sum_{i=3}^{i=4}\left(\sigma_i\, dW_i + \int_U q_i(u)y_{i-2} - \ln(1 + q_i(u))\tilde{N}(dt, du)\right),$$

with

$$LG_2 = \frac{1}{v_1}\frac{cu_2 v_2}{1 + \alpha u_2} - (\mu_3 + \theta) + \frac{\theta v_1}{v_2} - \mu_4 - \tau_2 v_2$$
$$+ \sum_{i=3}^{i=4}\left(\frac{\sigma_i^2}{2} + \int_U q_i(u) - \ln(1 + q_i(u))v(du)\right).$$

Therefore,

$$LG_2 \geq -\tau_2 v_2 - (\mu_3 + \theta + \mu_4) + \sum_{i=3}^{i=4}\frac{\sigma_i^2}{2} + 2N_2',$$

where $N_2' = \min\left\{\int_U \Pi_3(u) - \ln(1 + \Pi_3(u)), \int_U \Pi_4(u) - \ln(1 + \Pi_4(u))\right\}$, then

$$LG_2 \geq -\tau_2 v_2 + N_2$$

with

$$N_2 = \sum_{i=3}^{i=4}\frac{\sigma_i^2}{2} + 2N_2' - (\mu_3 + \theta + \mu_4).$$

Therefore,

$$\ln(v_1(t)v_2(t)) \geq \ln(v_1(0)v_2(0)) - \tau_2\int_0^t v_2(\eta)d\eta + N_2 t$$
$$+ \sum_{i=3}^{i=4}\int_0^t\left(\sigma_i\, dW_i(\eta) + \int_U q_i(u)y_{i-2} - \ln(1 + q_i(u))\tilde{N}(d\eta, du)\right).$$

Thus fact implies that

$$\frac{\ln(v_1(t)v_2(t))}{t} \geq \frac{\ln(v_1(0)v_2(0))}{t} - \tau_2\frac{1}{t}\int_0^t v_2(\eta)d\eta + N_2$$
$$+ \sum_{i=3}^{i=4}\frac{1}{t}\int_0^t\left(\sigma_i\, dW_i(\eta) + \int_U q_i(u)y_{i-2} - \ln(1 + q_i(u))\tilde{N}(d\eta, du)\right)$$

Using the strong law of large numbers for local martingales, we have

$$0 \geq -\tau_2 \frac{1}{t} \int_0^t v_2(\eta)d\eta + N_2,$$

then,

$$\frac{1}{t} \int_0^t v_2(\eta)d\eta \geq \frac{N_2}{\tau_2} \text{ a.s.}$$

The last equation of system (1) implies that

$$v_2(t) - v_2(0) \leq \theta \int_0^t v_1(\eta)d\eta - \mu_4 \int_0^t v_2(\eta)d\eta + \int_0^t \sigma_4 v_2(\eta)dW_4(\eta)$$
$$+ \int_0^t \int_U \Pi_4(u)v_2(\eta-)\tilde{N}(d\eta, du),$$

then,

$$\frac{v_2(t) - v_2(0)}{t} \leq \theta \frac{1}{t} \int_0^t v_1(\eta)d\eta - \mu_4 \frac{1}{t} \int_0^t v_2(\eta)d\eta + \frac{1}{t} \int_0^t \sigma_4 v_2(\eta)dW_4(\eta)$$
$$+ \frac{1}{t} \int_0^t \int_U \Pi_4(u)v_2(\eta-)\tilde{N}(d\eta, du),$$

so,

$$\mu_4 \frac{1}{t} \int_0^t v_2(\eta)d\eta + \frac{v_2(t) - v_2(0)}{t} \leq \theta \frac{1}{t} \int_0^t v_1(\eta)d\eta + \frac{1}{t} \int_0^t \sigma_4 v_2(\eta)dW_4(\eta)$$
$$+ \frac{1}{t} \int_0^t \int_U \Pi_4(u)v_2(\eta-)\tilde{N}(d\eta, du).$$

Since $\frac{1}{t} \int_0^t v_2(\eta)d\eta \geq \frac{N_2}{\tau_2}$, then

$$\frac{\mu_4 N_2}{\tau_2} + \frac{v_2(t) - v_2(0)}{t} \leq \theta \frac{1}{t} \int_0^t v_1(\eta)d\eta + \frac{1}{t} \int_0^t \sigma_4 v_2(\eta)dW_4(\eta)$$
$$+ \frac{1}{t} \int_0^t \int_U \Pi_4(u)v_2(\eta-)\tilde{N}(d\eta, du),$$

Using the strong law of large numbers for local martingales, we have

$$\frac{\mu_4 N_2}{\tau_2} \leq \theta \frac{1}{t} \int_0^t v_1(\eta)d\eta,$$

then,

$$\frac{1}{t} \int_0^t v_1(\eta)d\eta \geq \frac{\mu_4 K_2}{\theta \tau_2} \text{ a.s.}$$

$\square$

## 6. Numerical Simulations

Now, our main objective is to give numerical simulations of the model in (1); furthermore, we consider the equation

$$dY(t) = h(t, Y(t))dt + \sigma(t, Y(t))dW_t + \int_U \Pi(t, u)f(t-, Y(t-))\check{N}(dt, du). \qquad (10)$$

Then, the solution of (10) is

$$Y(t) = Y(0) + \underbrace{\int_0^t h(s, Y(s))ds + \int_0^t \sigma(s, Y(s))dW_s}_{\text{Part } I}$$

$$+ \underbrace{f(t-, Y(t-)) \int_0^t \int_U \Pi(s, u)\check{N}(ds, du)}_{\text{Part } II}. \qquad (11)$$

For approximation of part *I* in (10), we will use the Runge–Kutta method

$$Y_{k+1} = Y_k + \frac{1}{6}(K_1 + 2K_2 + 2K_3 + K_4)$$

where

$$K_1 = h(t_k, Y_k))\Delta t + \sigma(t_k, Y_k)(W_{k+1} - W_k),$$

$$K_2 = h(t_k + \frac{\Delta t}{2}t, Y_k + \frac{K_1}{2}))\Delta t + \sigma(t_k + \frac{\Delta t}{2}, Y_k + \frac{K_1}{2})(W_{k+1} - W_k),$$

$$K_3 = h(t_k + \frac{\Delta t}{2}, Y_k + \frac{K_2}{2}))\Delta t + \sigma(t_k + \frac{\Delta t}{2}, Y_k + \frac{K_2}{2})(W_{k+1} - W_k),$$

$$K_4 = h(t_k + \Delta t, Y_k + K_3))\Delta t + \sigma(t_k + \Delta t, Y_k + K_3)(W_{k+1} - W_k).$$

For part *II*, we will have two cases.
Consider any infinitesimal interval $[T_i, T_{i+1}) \subset (t_i, t_{i+1})$.
If there is no jump on this interval:

$$f(t-, Y(t-)) \int_{T_i}^{T_{i+1}} \int_U \Pi(s, u)\check{N}(ds, du) = 0.$$

If there is only one jump point $t_i \in [T_i, T_{i+1})$, so

$$f(t-, Y(t-)) \int_{T_i}^{T_{i+1}} \int_U \Pi(u)\check{N}(ds, du) = f(t_i-, Y(t_i-))\Pi(t_i, \zeta(t_i)),$$

then,

$$f(t-, Y(t-)) \int_0^t \int_U \Pi(s, u)\check{N}(ds, du) = \sum_{i=0}^{i=n} f(t_i-, Y(t_i-))\Pi(t_i, \zeta(t_i)).$$

Consequently, the Runge–Kutta method of problem (10) will be

$$Y_{k+1} = Y_k + \frac{1}{6}(K_1' + 2K_2' + 2K_3' + K_4')$$

where

$$K_1' = h(t_k, Y_k))\Delta t + \sigma(t_k, Y_k)(W_{k+1} - W_k)$$
$$+ f(t_k-, Y(t_k))\Pi(t_k, \zeta(t_k)),$$

$$K_2' = h(t_k + \frac{\Delta t}{2}t, Y_k + \frac{K_1'}{2}))\Delta t + \sigma(t_k + \frac{\Delta t}{2}, Y_k + \frac{K_1'}{2})(W_{k+1} - W_k)$$
$$+ f(t_k + \frac{\Delta t}{2}, Y(t_k) + \frac{K_1'}{2})\Pi(t_k + \frac{\Delta t}{2}, \zeta(t_k + \frac{\Delta t}{2})),$$

$$K_3' = h(t_k + \frac{\Delta t}{2}, Y_k + \frac{K_2'}{2}))\Delta t + \sigma(t_k + \frac{\Delta t}{2}, Y_k + \frac{K_2'}{2})(W_{k+1} - W_k)$$
$$+ f(t_k + \frac{\Delta t}{2}, Y(t_k) + \frac{K_2'}{2})\Pi(t_k + \frac{\Delta t}{2}, \zeta(t_k + \frac{\Delta t}{2})),$$

$$K_4' = h(t_k + \Delta t, Y_k + K_3'))\Delta t + \sigma(t_k + \Delta t, Y_k + K_3')(W_{k+1} - W_k)$$
$$+ f(t_k + \Delta t, Y(t_k) + K_3')\Pi(t_k + \Delta t, \zeta(t_k + \Delta t)).$$

We use the previous method to solve the system in (1) and the parameters values in Table 1.

**Table 1.** The values of the used parameters.

| Parameters | Figure 1 | Figure 2 | Figure 3 |
|---|---|---|---|
| $\delta$ | 0.7 | 0.1785 | 0.305 |
| $\tau_1$ | 0.92 | 0.37 | 0.47 |
| $c$ | 0.75 | 0.17 | 0.63 |
| $\kappa$ | 0.32 | 0.19 | 0.4 |
| $\theta$ | 0.53 | 0.048 | 0.048 |
| $\alpha$ | 0.84 | 0.037 | 0.037 |
| $\mu_1$ | 0.91 | 0.0342 | 0.034 |
| $\mu_2$ | 0.532 | 0.0401 | 0.0401 |
| $\mu_3$ | 0.483 | 0.0481 | 0.058 |
| $\mu_4$ | 0.489 | 0.0432 | 0.0432 |
| $\sigma_1$ | $10^{-5}$ | $10^{-4}$ | $10^{-4}$ |
| $\sigma_2$ | $3 \times 10^{-5}$ | $3 \times 10^{-5}$ | $3 \times 10^{-5}$ |
| $\sigma_3$ | $3 \times 10^{-5}$ | $3 \times 10^{-5}$ | $3 \times 10^{-5}$ |
| $\sigma_4$ | $3 \times 10^{-5}$ | $3 \times 10^{-5}$ | $3 \times 10^{-5}$ |
| $\Pi_1(u)$ | $-0.051$ | $-0.051$ | $-0.052$ |
| $\Pi_2(u)$ | $-0.007$ | $-0.007$ | $-0.007$ |
| $\Pi_3(u)$ | $-0.009$ | $-0.006$ | $-0.007$ |
| $\Pi_4(u)$ | $-0.009$ | $0.008$ | $0.007$ |

In Figure 1, we show the dynamics of $u_1$, $u_2$, $v_1$, and $v_2$, and we observe that both free jobs and the total labor force in the deterministic and the stochastic curves tend to zero; furthermore, we have the extinction of free jobs and the total labor force.

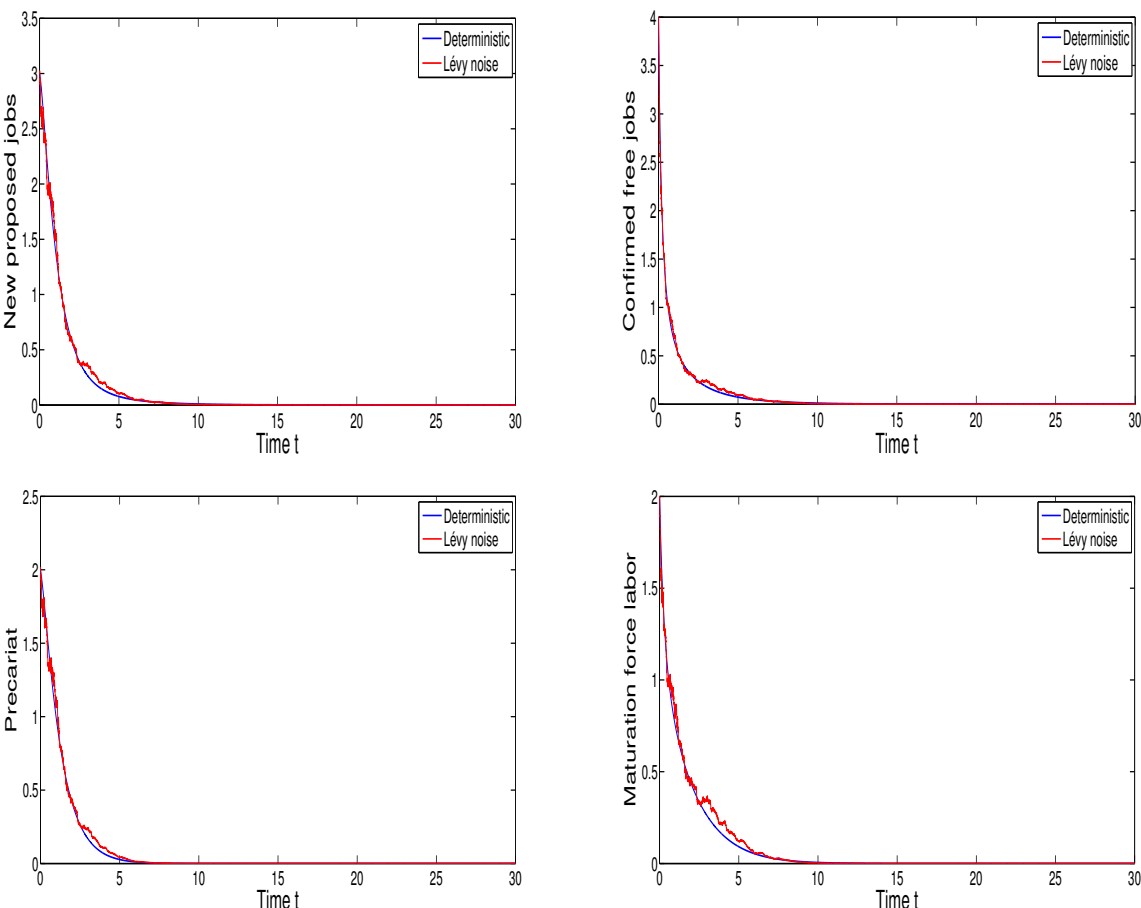

**Figure 1.** The behavior of free jobs and the labor force.

Figure 2 represents the behavior of $u_1$, $u_2$, $v_1$, and $v_2$. We remark that the labor force is extinct and the free jobs stay non negative; this means that the population of the labor force will be extinct and the free jobs will persist.

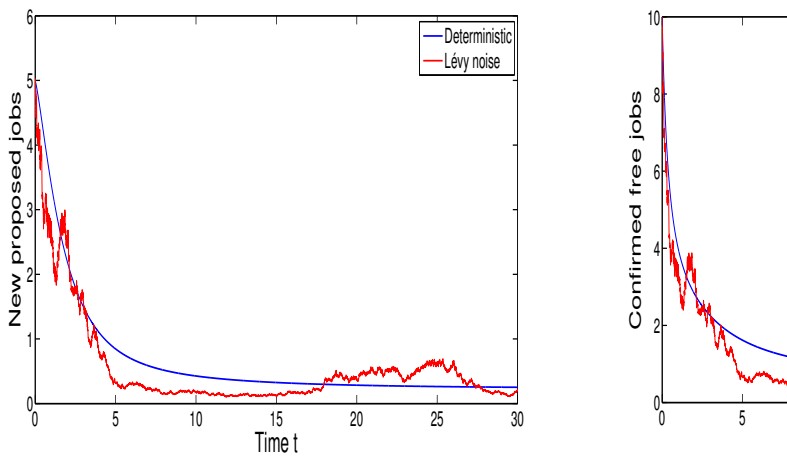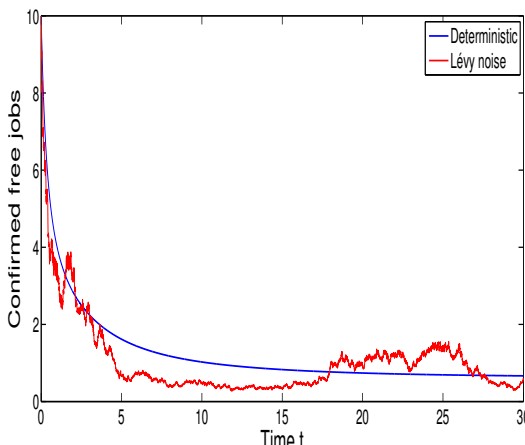

**Figure 2.** *Cont.*

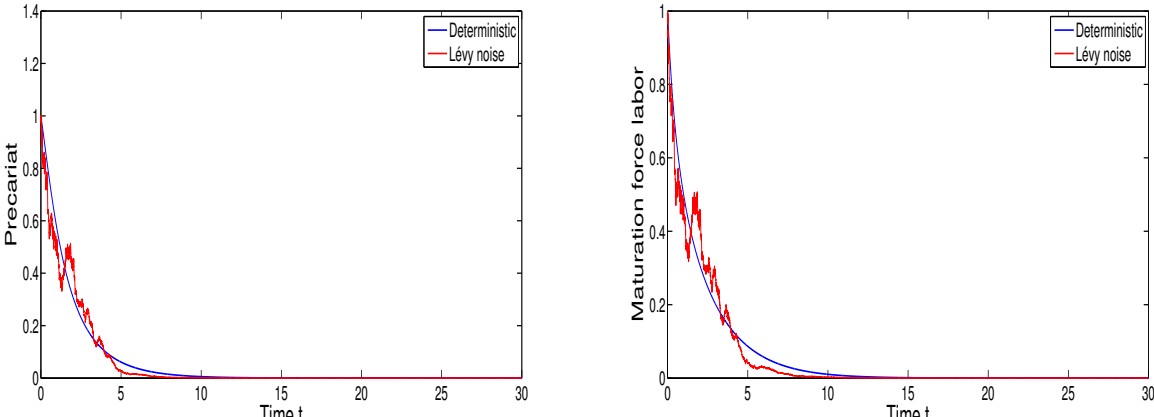

**Figure 2.** The behavior of free jobs and the labor force.

In Figure 3, we illustrate the interaction between $u_1$, $u_2$, $v_1$, and $v_2$. We show that both free jobs and and the labor force stay strictly positive in the two cases of stochastic and deterministic, then we have the persistence of both the free jobs and the labor force.

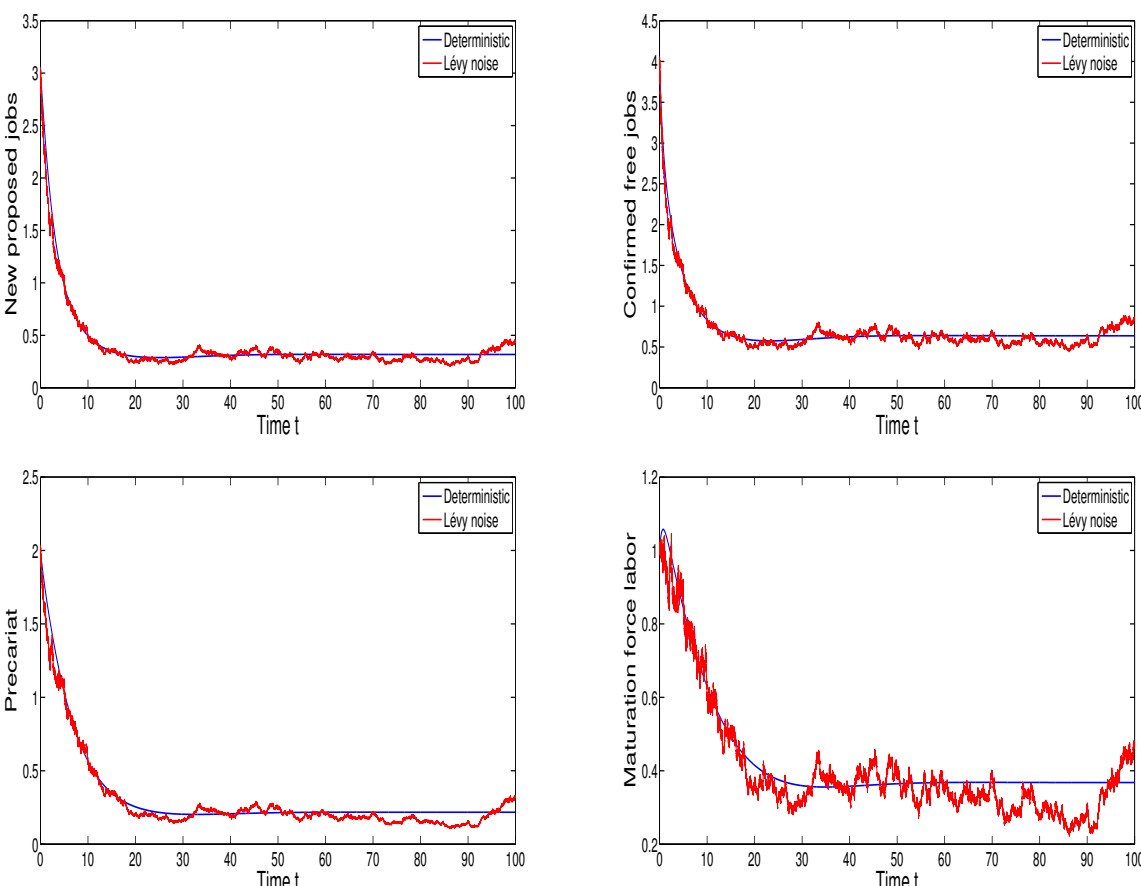

**Figure 3.** The behavior of free jobs and the labor force.

## 7. Conclusions and Discussion

In this paper, we have studied the interaction between free jobs and the labor force by decomposing each class into two subclasses. For the free jobs, we have considered the proposed new free jobs by the recruiting company and the confirmed free jobs. For the labor force, in order to account for labor force seasonality or internship periods, we divide

the labor force into the precariat population and the mature labor force. Moreover, we have studied the well-posedness of the solution, by proving the existence, uniqueness, and the stochastic ultimate boundedness. In addition, we examined the behavior of our model in three cases. The first of them concerns the extinction of both free jobs and the labor force; the second concerns the extinction of the labor force and the persistence of free jobs; and the last concerns the persistence of both free jobs and the labor force. Finally, our study is supported by a numerical simulations in order to validate our theoretical findings. In a possible follow-up to our investigation in this article, we can extend problem (1) to different fractional derivatives, which rely on several fractional derivative operators, such as the Liouville–Caputo, Riemann–Liouville, and other fractional derivative operators [24,25]. In addition, we can extend our system to an impulsive problem such as that studied in [26].

**Funding:** This research received no external funding.

**Conflicts of Interest:** This work does not have any conflicts of interest.

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
