# Peer review of "Stochastic Capital–Labor Lévy Jump Model with the Precariat Labor Force"

_mca, doi:10.3390/mca27060093_

Round 1

Reviewer 1 Report

The peer-review paper has an interesting content, clear presentation of the main body and can be continued in the future research.

At the same time, there are a number of debatable points:

1. The author has ignored the requirements of the Research Manuscript Sections. It should be restructured according to the requirements.

2. Line 32 - Link to the reference isn't correct: in [24], Riad et al.

3. Lines 21, 27 - It's better to fix the link to the sources with the indication of the authors (In [21], In [23]).

4. Lines 37-54 - This information should be transferred to the created Materials and Methods section. You should also add the initial information used for further testing of the author's hypothesis.

5. The Literature Review Section is missed in this paper.

6. For the relevant use of mathematical equations in other fields of knowledge (physics, economics, etc.), it is necessary that the dimensionality of the components of the equation in its right and left parts should be the same.

Let us analyze the system of equations (1) from this point of view.

Obviously, in the first and second equations the sum

must have a dimensionality similar to  and   and, at the same time, the dimensionality of the numerator  must be the square of the dimensionality of the denominator, and this is impossible, since αu_2 is added to the dimensionless number (1) in the denominator. At the same time, the dimensionality   (the competition rate between confirmed free jobs) should be the inverse of the values  and , since it is a factor of (). And this is not confirmed by your definition of the specified values.

This raises doubts about the use of all subsequent mathematical transformations of this system of equations and conclusions after their use.

Author Response

Author response to referees remarks to the paper

Stochastic Capital-labor Lévy jump model with  the precariat labor force

We are very grateful to the referees for their valuable suggestions. All of them are taken into account in the revised version of the paper. Detailed responses are given below.

Reviewer 1

  1. Line 32 - Link to the reference isn't correct: in [24], Riad et al.

Thank you for this remark, we have corrected.

  1. Lines 21, 27 - It's better to fix the link to the sources with the indication of the authors (In [21], In [23]).

       It’s corrected.

  1. Lines 37-54 - This information should be transferred to the created Materials and Methods section. You should also add the initial information used for further testing of the author's hypothesis.

In this paper our objective is to give a mathematical study applied to the economy, this obliges us to illustrate mathematical results and to validate them by numerical simulations. That’s it’s possible by the journal you can see for example the paper “https://doi.org/10.3390/mca27060091

  1. The Literature Review Section is missed in this paper.

   In the introduction we have given the literature of our study and since our main objective it’s to use the applied mathematics in the economies. That’s it’s possible by the journal you can see for example the paper “https://doi.org/10.3390/mca27060091

  1. For the relevant use of mathematical equations in other fields of knowledge (physics, economics, etc.), it is necessary that the dimensionality of the components of the equation in its right and left parts should be the same.

For example the unit of τ1 is the [densities]-1 [time] -1 the same remark for τ2. We note also our study is devoted to economies problem more precisely the capital-labor system.

Reviewer 2 Report

Review manuscript ID: mca-2000956
Type of manuscript: Article
Title: Stochastic Capital-labor Lévy jump model with  the precariat labor force
Author: Jaouad Danane
Journal: Mathematical and Computational Applications
Special Issue: Ghana Numerical Analysis Day
Date: 31 October 2022

The manuscript describes the mathematical and stochastical analysis of free jobs and labor force model, as well the numerical study of such mathematical model. Here are some remarks.

* Line 5: we also show ....

* Line 23: showed

* Line 25: explained

* Line 30: capital

* Line 32: gave

* Line 40: spacing needed

* Line 53: comma needed

* Line 55: as follows:

* a.s. should be explained

* Page 2 bottom: square bracked should be enlarged.

* Page 2 bottom: as follows: (colon needed)

* Line 70: we prove

* Line 71: i.e., (comma needed)

* Use either "Ito's formula" or "the Ito formula", but not both.

* Line 77: insufficient.

* References regarding Ito's formula and local martingales are ought to be provided.

* the Runge-Kutta method

* Page 18: Figure 2 represents ... .

* All figures are poor quality, they should be improved, particularly the inconsistent fontsize of labeling.

* Line 120: three cases

* Line 120: our model. The first of ...

* English should be corrected before resubmission.

Author Response

Author response to referees remarks to the paper

Stochastic Capital-labor Lévy jump model with  the precariat labor force

We are very grateful to the referees for their valuable suggestions. All of them are taken into account in the revised version of the paper. Detailed responses are given below.

 Reviewer 2

Line 5: we also show ....

* Line 23: showed

* Line 25: explained

* Line 30: capital

* Line 32: gave

* Line 40: spacing needed

* Line 53: comma needed

* Line 55: as follows:

* a.s. should be explained

* Page 2 bottom: square bracked should be enlarged.

* Page 2 bottom: as follows: (colon needed)

* Line 70: we prove

* Line 71: i.e., (comma needed)

* Use either "Ito's formula" or "the Ito formula", but not both.

* Line 77: insufficient.

* References regarding Ito's formula and local martingales are ought to be provided.

* the Runge-Kutta method

* Page 18: Figure 2 represents ... .

* All figures are poor quality, they should be improved, particularly the inconsistent fontsize of labeling.

* Line 120: three cases

* Line 120: our model. The first of ...

Thank you for all your suggested valuable remarks and comment, all them are added to the revised version of our paper.

Round 2

Reviewer 1 Report

Dear author, I recommend this paper for open publication.

The article can be published in the current version.

Good luck!